# Interleukin-33 regulates the endoplasmic reticulum stress of human myometrium via an influx of calcium during initiation of labor

Li Chen[1]*[†], Zhenzhen Song[1][†], Xiaowan Cao[2][†], Mingsong Fan[1], Yan Zhou[1], Guoying Zhang[1]*

[1]Department of Obstetrics, The First Affiliated Hospital of Nanjing Medical University, Nanjing, China; [2]Department of Obstetrics, The Affiliated Wuxi Maternity and Child Health Care Hospital of Nanjing Medical University, Wuxi, China

## Abstract

**Background:** Inflammation is currently recognized as one of the major causes of premature delivery. As a member of the interleukin-1β (IL-1β) family, interleukin-33 (IL-33) has been shown to be involved in normal pregnancy as well as a variety of pregnancy-related disorder. This study aims to investigate the potential function of IL-33 in uterine smooth muscle cells during labor.

**Methods:** Myometrium samples from term pregnant (≥37 weeks gestation) women were either frozen or cells were isolated and cultured. Immunohistochemistry and western blotting were used to assess the distribution of IL-33. Cultured cells were incubated with lipopolysaccharide (LPS) to mimic inflammation as well as in the presence of 4μ8C (IRE1 inhibitor III) to block endoplasmic reticulum (ER) stress and BAPTA-AM, a calcium chelator.

**Results:** LPS reduced the expression of nuclear IL-33 in a time-limited manner and induced ER stress. However, knockdown of IL-33 increased LPS-induced calcium concentration, ER stress and phosphorylation of nuclear factor kappa-B (NF-κB), and P38 mitogen-activated protein kinase (P38 MAPK). In addition, siRNA IL-33 further stimulates LPS enhanced cyclooxygenase-2 (COX-2) expression via NF-κB and p38 pathways. IL-33 expression was decreased in the nucleus with the onset of labor. LPS-induced ER stress and increased expression of the labor-associated gene, COX-2, as well as IL-6 and IL-8 in cultured myometrial cells. IL-33 also increased COX-2 expression, but after it was knocked down, the stimulating effect of LPS on calcium was enhanced. 4μ8C also inhibited the expression of COX-2 markedly. The expression of calcium channels on the membrane and intracellular free calcium ion were both increased which was accompanied by phosphorylated NF-κB and p38.

**Conclusions:** These data suggest that IL-33 may be involved in the initiation of labor by leading to stress of the ER via an influx of calcium ions in human uterine smooth muscle cells.

**Funding:** This study was supported by grants from the National Natural Science Foundation of China (No. 81300507).

## Editor's evaluation

This paper addresses an interesting and important topic bearing on the initiation of labor at the end of pregnancy. It tests a hypothesis invoking interleukin-33 in uterine smooth muscle in the third trimester of pregnancy in an endoplasmic reticular stress. This in turn results in an alteration of $ca^{2+}$ homeostasis that might be involved in initiating labor. The study was made in human myometrial cells enhancing its clinical translatability.

*For correspondence:
jasmine_lichen@sina.cn (LC);
zhangguoying@jsph.org.cn (GZ)

[†]These authors contributed equally to this work

Competing interest: The authors declare that no competing interests exist.

## Introduction

Preterm delivery occurs before 37 weeks of gestation and is one of the major causes of perinatal morbidity and mortality. The main difference between preterm labor (PTL) and term labor (TL) is when the initiation of labor begins. It is widely accepted that infection, uterine over-distension, decline in progesterone action, breakdown of maternal–fetal tolerance, stress, and other unknown reasons are involved in PTL (Romero et al., 2014). However, all of these reasons lead to inflammatory events. These are mainly due to the identification of microorganisms and their products, and increased secretion of proinflammatory cytokines and chemokines in the decidua and myometrium, which enable leukocytes to infiltrate the uterus leading to its activation (Goldenberg et al., 2008; Lamont, 2015).

As a member of IL-1 family, IL-1β is one of the well-recognized inflammatory factors that induce the initiation of labor. Interleukin-33 (IL-33) is another member of the IL-1 family and it is expressed in a variety of cells, including innate immune (macrophage and dendritic), Th2, B, epithelial, endothelial, and muscle cells. Also, it has been shown to have a key role in various diseases, including inflammation-associated autoimmune (including asthma), cardiovascular, and allergic diseases (Cayrol and Girard, 2018; Drake and Kita, 2017).

Recently, Huang et al., 2017 found that after exogenous administration of IL-33, it was able to activate B cell-mediated immune regulation and promote the expression of progesterone-induced blocking factor 1 (PIBF1), which plays an essential role in safeguarding against PTL. Preliminary data showed that IL-33 was mainly located in the nuclei of myometrial cells, and with the onset of labor, IL-33 expression in the nucleus decreased. Studies also showed that mast cells responded to the secretion of IL-33 after exposure to the allergens and this affected airway smooth muscle hyper-responsiveness by promoting the contraction and enhancing calcium influx of airway smooth muscle, leading asthma attacks (Bønnelykke et al., 2014; Kaur et al., 2015). In addition, IL-33 was found to be effective in alleviating the myocardial endoplasmic reticulum (ER) stress response caused by myocardial injury (Yao et al., 2017; Theoharides et al., 2010). These studies suggested a potential role of IL-33 in human parturition and prompted us to investigate the changes in IL-33 expression in the nuclei of the myometrium during the third trimester of pregnancy.

## Materials and methods

### Specimen collection

Patients were separated into four groups: (1) TL: samples were obtained from women who had the presence of contractions of sufficient strength and frequency to effect progressive effacement and dilation of the cervix and were between 37 and 41 + 6 weeks' gestation; (2) PTL: samples were obtained from women who had the presence of contractions of sufficient strength and frequency to effect progressive effacement and dilation of the cervix and were between 28 and 36 + 6 weeks' gestation; (3) term nonlabor (TNL): these samples were obtained from women who were between 37 and 41 + 6 weeks' gestation without any signs of onset of labor; (4) preterm nonlabor (PNL): these samples were obtained from women who were between 28 and 36 + 6 weeks' gestation without any sign of onset of labor.

When undergoing cesarean section, a sample of the uterine smooth muscle ($1.0 \times 1.0 \times 1.0$ cm$^3$) was cut from the site of incision at the lower and upper edges of the uterus. The muscle was washed three times in precooled phosphate-buffered saline (PBS) to remove excess blood. Then these were quickly frozen in liquid nitrogen, and stored at −80°C. In some cases, similar sized samples were stored in cold PBS and used for in vitro cultures of myometrial smooth muscle cells.

### Western blotting

Proteins were extracted from human primary myometrial cells and human uterine smooth muscle tissues. Proteins were separated by using 8% or 10% sodium dodecyl sulfate–polyacrylamide gel electrophoresis, and these were transferred to polyvinylidene fluoride membranes (IPVH00010, Mreck Millipore). Nonspecific binding sites were blocked by incubation with 5% nonfat milk for 2 hr. Membranes were then incubated with specific antibodies overnight at 4°C and washed three times with phosphate-buffered saline with Tween-20 (PBST) and then incubated with infrared dye-labeled secondary antibodies for 2 hr at room temperature.

**Table 1.** The following primers were used in this study.

| Target genes | Forward primer | Reverse primer |
|---|---|---|
| IL-33 | TGCCAACAACAAGGAACACTCTG | CACTCCAGGATCAGTCTTGCATTC |
| XBP1s | CTGAGTCCGCAGCAGGTG | GTCCAGAATGCCCAACAGGA |
| GRP78 | CCCGTCCAGAAAGTGTTG | CAGCACCATACGCTACAG |
| β-Actin | CTCCATCCTGGCCTCGCTGT | GCTGTCACCTTCACCGTTCC |

After further washes with PBST, membranes were visualized using a Bio-Rad gel imager and the results were normalized to β-actin or GAPDH protein and expressed in arbitrary units. The dilutions of the antibodies used were as follows: anti-IL-33 antibody (1:1000, ab54385, Abcam), anti-p-IRE1α antibody (1:1000, ab124945, Abcam), anti-XBP1s antibody (1:1000, 83418, Cell Signaling Technology), anti-GRP78 antibody (1:1000, 11587-1-AP, Proteintech), anti-COX-2 antibody (1:1000, ab179800, Abcam), anti-p38 antibody (1:1000, D13E1, 8690, Cell Signaling Technology), anti-NF-κB antibody (1:1000, D14E12, 8242, Cell Signaling Technology), anti-phospho-p38 antibody (1:1000, D3F9, 4511, Cell Signaling Technology), anti-phospho-NF-κB antibody (1:1000, 93H1, 3033, Cell Signaling Technology), anti-Cav 3.1 antibody (1:2000, DF10014, Affinity), and anti-Cav 3.2 antibody (1:500, sc-377510, Santa Cruz Biotechnology).

## Immunohistochemistry

The fixed paraffin-embedded sections of human myometrium were rehydrated in a graded series of decreasing alcohol concentrations. Sections were incubated in 3% hydrogen peroxide for 30 min to block endogenous peroxidase and 2% normal goat serum for 1 hr to reduce nonspecific binding. Then samples were incubated overnight at 4°C with goat anti-human IL-33 monoclonal antibody (MAB36253, R&D) at a concentration of 0.1 μg/ml and then one to three drops of biotinylated secondary antibodies for 2 hr at room temperature. One to five drops of DAB chromogen solution were added so as to cover the entire tissue section and incubated for 10 min. The sections were rinsed in deionized H$_2$O and then the slides were drained. The stained tissues were covered with a coverslip of an appropriate size and these were then visualized under a microscope using a bright-field illumination.

## Quantitative real-time PCR

Total RNA was isolated using Trizol reagent (SN114, NanJing SunShine Biotechnology). The RNA was reverse transcribed to cDNA using a cDNA first strand synthesis kit (D6110A, Takara Bio), according to the manufacturer's instructions. The upstream and downstream primer sequences used are shown in *Table 1*. Quantitative real-time PCR was then conducted with SsfastTMEvaGreen Supermix (172-5201AP, Bio-Rad) in a Bio-Rad CFX96 Touch Real-Time PCR. The relative abundance of mRNA in the samples was normalized to that of β-actin using the comparative cycle threshold method ($2^{-\Delta\Delta CT}$).

## Immunofluorescence

Tissue sections and cells were fixed with 4% paraformaldehyde in PBS for 30 min at room temperature, washed three times with PBS and repaired with 75% glycine for 10 min. Then they were permeabilized with 0.5% Triton X-100 in PBS for 30 min, and then washed three times with PBS and blocked with 3% bovine serum albumin (BSA, sh30087.02, HyClone) in PBS for 1 hr. These operations were conducted at room temperature. Primary antibody incubation was performed overnight at 4°C with goat monoclonal anti-IL-33 antibody (1:200, MAB36253, R&D) diluted in 0.2% BSA in PBS. For detection, the tissues were incubated for 2 hr at room temperature with Alexa Flour 594 secondary antibody (1:500, A32758, Life Technologies) which was diluted in 0.2% BSA in PBS. The samples were washed as described above, mounted using Antifade Mounting Medium with 4',6-diamidino-2-phenylindole (DAPI) (P0131, Beyotime Biotechnology) and analyzed using a confocal fluorescence microscope (LSM 800, Zeiss).

## Human myometrial cell culture

The full thickness of uterine samples was obtained from women undergoing a scheduled term (≥37 weeks of gestation) cesarean section delivery. Patients who lacked signs of infection or known

pregnancy complications and were clinically defined as not in labor on the basis of intact membranes, a closed cervix and a quiescent uterus. After delivery of the placenta, samples of myometrium (1 cm$^3$ ) were excised from the upper incisional margin of the lower uterine segment, and these were immediately washed in ice-cold PBS. Myometrium was carefully minced into small pieces of about 1 mm$^3$, subsequently washed and incubated with gentle agitation for 40 min, at 37°C, with collagenase IA (C9891, Sigma-Aldrich) and collagenase XI (C7657, Sigma-Aldrich) each at 0.5 mg/ml in Dulbecco's modified Eagle medium (DMEM)/F-12 media (A4192001, Gibco) supplemented with BSA (sh30087.02, HyClone) at 1 mg/ml. The dispersed cells were separated from nondigested tissue by filtration through a cell strainer (70 μm, 431751, Corning), and then collected by centrifugation of the filtrate at 3000 rpm for 5 min at room temperature. The cells were suspended in DMEM/F-12 media supplemented with 10% fetal bovine serum (10270-106, Gibco) and 1% penicillin/streptomycin/ amphoterin B (15240062, Gibco). Cells were then dispersed in 12.5-cm$^2$ plastic culture flasks (12631, JETBIOFIL) or on glass coverslips into 12-well plates (7516, JETBIOFIL). Culture medium was changed every 48 hr. Cells were maintained at 37°C in a humidified atmosphere (5% $CO_2$ in air) until they were semi-confluent (usually 9 days after plating) after which they were detached using 0.25% trypsin–Ethylene Diamine Tetraacetic Acid (EDTA) (15050065, Gibco). Experiments were performed using cells between passages 1 and 4. Culture media were collected after incubations and the protein concentration of multiple secreted cytokines was determined by enzyme-linked immunosorbent assay (ELISA).

## Cell interference

In order to elucidate the involvement of IL-33 in the regulation of uterine contractions during delivery, knockdown experiments were performed in myometrial cells. After starving the cells for 2 hr in serum-free medium, a Lipofectamine 2000 (11668027, Thermo Fisher Scientific) and siRNA IL-33 (20 μM, Gene Pharma) mixture was added to the cells according to the manufacturer's instructions. After incubation for 6 hr, the medium was replaced for a further 48 hr and the efficiency of siRNA transfection was measured by western blotting analysis.

## Medicine loading

Some cells were incubated with lipopolysaccharides (LPS; L8880, Solarbio). 10 μg/ml of LPS was added to treat the cells for 5, 10, 15, and 30 min as well as for 1, 3, 6, 9, 12, 18, and 24 hr, respectively, whereas the untreated uterine smooth muscle cells were used as a control group.

4μ8C (80 μM, HY-19707, Master of Small Molecules) was used to blocked ER stress and BAPTA-AM (25 μg of MAB120503, Abcam), a membrane-permeable calcium ion chelator was used to block the action of cytoplasmic calcium in the cells. These were added to the cells 1 hr before LPS treatment.

## Enzyme-linked immunosorbent assays

Sandwich ELISAs were used to determine the cytokine concentrations in culture medium supernatants following various treatments. ELISA kits for human IL-6 and IL-8 were purchased from Meimian Biotechnology (Yancheng, Jiangsu, China; MM-0049H1 and MM-1558H1, respectively). Culture media were diluted 1:5 for IL-6 and IL-8 determinations using diluent solutions supplied by the manufacturer to ensure the absorbance readings would remain within the ranges of the standard curves. Absorbance readings were conducted using μQuantTM software (128C-400, BioTek Instruments) according to the manufacturer's instructions.

## Statistical analysis

The data are presented as mean percentages of the control ± standard deviation (SD). The data from western blots are presented as means ± standard error of the mean (SEM) of the ratios. Statistical analyses were compared using an unpaired two-tailed Student's $t$ test or a one-way analysis of variance and Bonferroni multiple comparisons test with SPSS software (version 25.0 for Windows, IBM Inc, Chicago, IL, USA) and the figures were generated using GraphPad Prism.

# Results

## Nuclear expression of IL-33 is reduced during labor

In order to establish the nuclear function of IL-33, we first confirmed the presence of the nuclear protein in human uterine smooth muscle cells. Immunohistochemistry with the anti-IL-33 antibody revealed that IL-33 was expressed in the nuclei of myometrial cells. The TL and PTL groups showed weak staining of IL-33 in the nuclei of myometrium cells. In contrast, strong nuclear staining of IL-33 was observed in the myometrial cells of TNL and PNL groups (*Figure 1A*).

In order to identify the localization of IL-33 changes during labor, immunofluorescence staining was performed on myometrial sections. This also confirmed the nuclear localization of IL-33 in the myometrial cells. It should be noted that the expression of nuclear IL-33 was more intense in the TNL and PNL groups when compared with TL and PTL groups. There were similar levels of staining in the two no labor groups as well as the two labor groups (*Figure 1B*). Western blotting analysis revealed that there was no difference in the expression of total IL-33 among all groups. To confirm the nuclear localization of the IL-33, we separated the cytoplasmic and nuclear fraction of myometrial cells. There were decreased levels of nuclear IL-33 from the TNL to the TL groups, which was consistent with the change seem from PNL to PTL. The results of qPCR suggested that mRNA expression of total IL-33 was not different between the different groups (*Figure 1C*).

## Treatment with LPS resulted in a time-dependent nuclear IL-33 decrease

To examine the function and the mechanisms of nuclear IL-33 during parturition, myometrial cells were incubated with LPS and the kinetic changes of IL-33 were visualized by confocal microscopy. Following treatment with LPS, there was a marked decrease in endonuclear IL-33 levels for approximately 3 hr and then this increased to above control values by 12 hr, which was higher than that in the control group (*Figure 2A*). For the following 12 hr, the levels appeared to decrease again. We next investigated the effect of LPS on the localization of IL-33 in these cells. Western blots confirmed that the expression levels of total IL-33 were reduced within 6 hr, and then peaked at 12 hr. The cytoplasmic and nuclear fractions of primary myometrial cells were also isolated at different time points after LPS treatment. Western blot analysis showed that nuclear IL-33 reduced with the stimulation by LPS. However, the levels peaked again in the nucli at 12 hr, and were higher than seen in control cells (*Figure 2B*). This indicates that LPS induces a time-dependent reduction of IL-33 in the nucleus.

## Silencing of IL-33 enhances LPS-induced calcium ion levels

To investigate whether the presence of IL-33 regulates uterine smooth muscle contraction, knockdown experiments targeting IL-33 were performed before LPS treatment for 0.5, 1, 3, and 6 hr, respectively. After stimulating for 0.5–6 hr, an increase in intracellular calcium levels was observed by using laser confocal microscopy to observe the added intracellular fluorescent probes which were consistent with the calcium channels status (*Figure 3A*). Apparent enhancement in the protein levels of calcium channels proteins, Cav3.1 and Cav3.2, was also demonstrated after siRNA-based LPS treatment while siRNA-mediated efficient knockdown of IL-33 was simultaneously observed (*Figure 3B*).

## Inhibition of cytoplasmic calcium attenuates LPS-induced ER and cyclooxygenase-2

LPS was able to induce the levels of cytoplasmic calcium as well as ER stress in uterine smooth muscle cells, but whether there was a correlation between the two parameters was not known. In this study, BAPTA-AM was used to evaluate the role of elevated cytoplasmic calcium levels during ER stress. The results showed an attenuation in the expression of LPS-induced cyclooxygenase-2 (COX-2) after 12 hr of preloading cells with BAPTA-AM (*Figure 4A*). After LPS stimulation for 6 and 12 hr, respectively, the intracellular ER state was assessed by western blotting analysis. We found that p-IRE1α and XBP1s in the myocytes of prechelated-based stimulation showed a decreasing trend when compared with cells stimulated by LPS directly. Furthermore, on the basis that there was no significant difference detected after LPS treatment when directly compared to controls, there was no apparent alteration in protein levels of GRP78 in prechelated-based stimulation (*Figure 4B*).

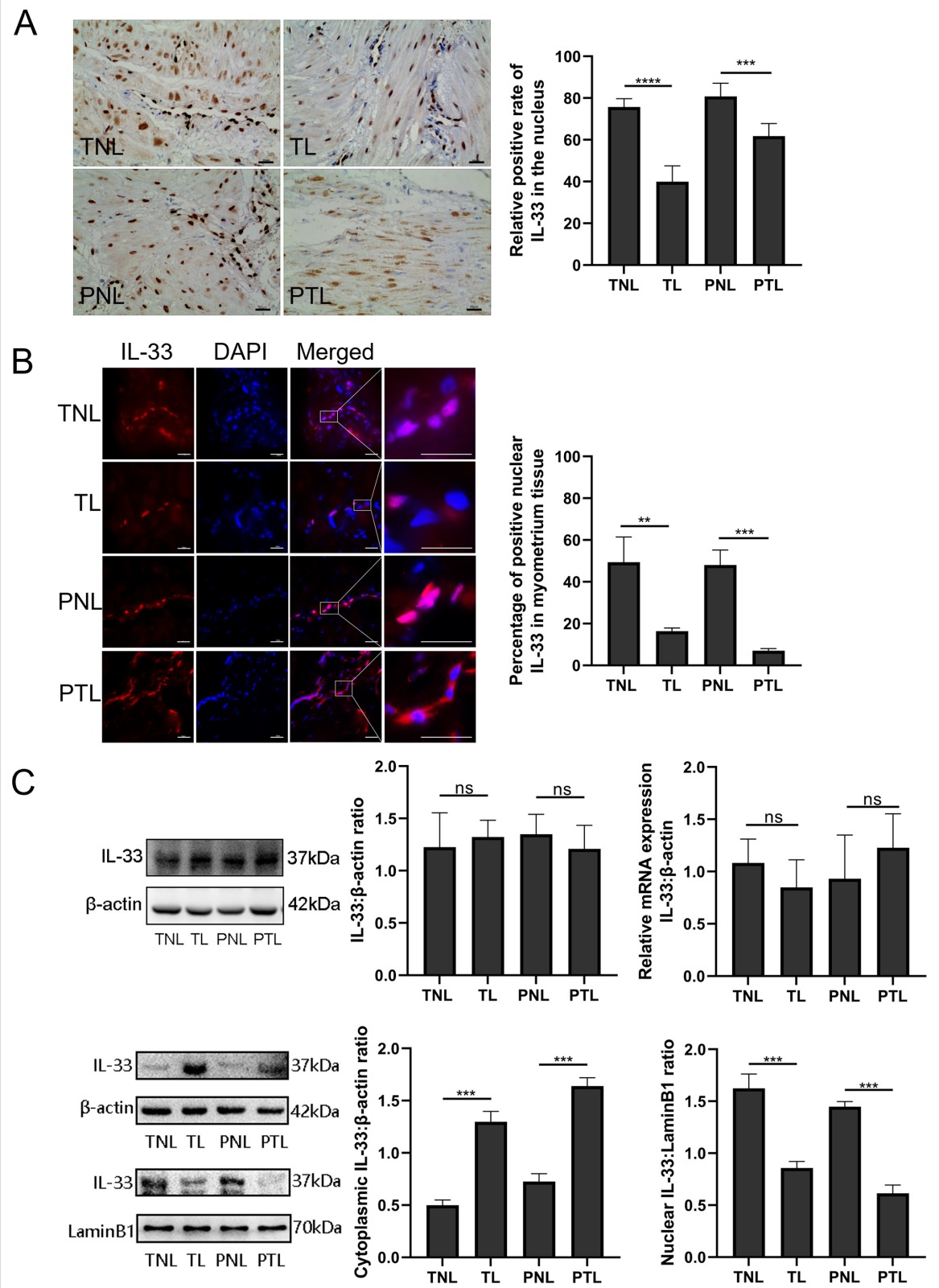

**Figure 1.** Nuclear localization of the interleukin-33 (IL-33) and reduced level associated with the onset of labor. The fixed paraffin-embedded sections of human myometrium were immune stained with anti-IL-33 antibody. (**A**) Immunohistochemistry results showing that in term nonlabor (TNL) and preterm nonlabor (PNL) tissues, IL-33 was mostly located in the nuclei which reduced sharply and emerged in the cytoplasm with the initiation of labor as shown in these representative images of term labor (TL) and preterm labor (PTL) samples. Quantification of IL-33 expression within the nuclear

*Figure 1 continued on next page*

Figure 1 continued

regions showed apparent differences between labor and nonlabor tissues (*n* = 8). (**B**) The sections of human myometrium were visualized using an Alexa Flour 594 secondary antibody labeled with IL-33 (red), and the nuclei were stained with DAPI (blue). Analysis of nuclei was performed by using confocal microscopy and the fluorescence signals of IL-33 and nuclei were superimposed. Representative immunofluorescence photomicrographs reflected this same phenomenon. (**C**) From western blots, we also can see that the nonlabor groups had more nuclear expression and less cytoplasmic expression of IL-33 compared to labor groups while the total IL-33 had no obvious differences between groups. Quantitative real-time PCR (qRT-PCR) discovered no apparent alteration in the levels of IL-33 mRNA (*n* = 3). Representative blots are shown. The data are presented as the mean percentages of control ± standard deviation (SD). The data from western blots and immunofluorescence are presented as the means ± standard error of the mean (SEM) of the ratios. Statistical analyses were compared using an unpaired two-tailed Student's *t* test. \*\*p＜0.01, \*\*\*p＜0.001, \*\*\*\*P ＜0.0001 when compared to control; bar, 50 µm.

The online version of this article includes the following source data for figure 1:

**Source data 1.** Nuclear localization of the IL-33 and reduced level associated with the onset of labor.

## IL-33 affects the LPS-induced ER stress response in myometrium cells

Since studies have indicated a crucial role of ER stress during successful pregnancies as well as in the pathogenesis of preeclampsia, we decided to assess whether the ER stress response also played a role in labor. As shown in *Figure 5A*, the protein expression of the ER stress-sensing molecules, p-IRE1α and XBP1s, was significantly increased in the TL and PTL groups when compared with the TNL and PNL groups. However, there was no significant change in GRP78 protein expression among the groups.

In order to verify whether the transcription levels changed, qPCR results were compared those of western blotting analysis and were found to be similar (*Figure 5A, B*). qPCR showed that the mRNA expression of GRP78 had no intergroup differences. However, XBP1s mRNA expression was increased in PTL when compared with PNL, but there was no difference between the TNL and TL groups. Overall, these findings support the hypothesis that the activation of ER stress is a potential mechanism underlying labor. In order to confirm this, we stimulated primary myometrial cells with LPS and induced the ER stress response without altering GRP78 protein levels. We found that the expression levels of p-IRE1α and XBP1s reached their peaks at 30 min and 1 hr, and then decreased to almost the basal levels at 12 hr (*Figure 5C*). We next suppressed the expression with IL-33 siRNA in these cells and found that IL-33 knockdown heightened the LPS-induced ER stress response. This is shown in *Figure 5D* and in contrast to the LPS group, XBP1s and phosphorylated IRE1α protein levels were increased in the IL-33 siRNA group.

## IL-33 siRNA and ER stress response regulated LPS-induced COX-2 expression in myometrial cells

So as to determine whether IL-33 had a direct impact on expression of COX-2, we introduced IL-33 siRNA into primary myometrial cells and treated them with LPS for 12 hr. COX-2 was upregulated of in the IL-33 siRNA group (*Figure 6A*). The previous results showed that a milder ER stress response occurred with the onset of labor and this could be modulated by cytoplasmic calcium levels. To define the possible effect of the ER stress during the labor, an IRE1 inhibitor, 4µ8c, was used to block the LPS-induced ER stress response. Western blots were conducted to examine whether the expression of COX-2 could be inhibited. The results demonstrated that 4µ8C inhibited the expression of COX-2 markedly (*Figure 6B*).

## IL-33 may affect LPS-induced COX-2 expression via p38/mitogen-activated protein kinase and nuclear factor kappa-B signaling pathways

We also found the expression of phosphorylation of p38 /mitogen-activated protein kinase (MAPK) and nuclear factor kappa-B (NF-κB) increased significantly when LPS caused an induction of COX-2 expression, and these peaked after 30 min (*Figure 7A*). Based on the above observations, we wondered whether IL-33 was involved in these signaling pathways, thereby affecting the expression of COX-2. Based on the knockdown experiments targeting IL-33 which showed that when siRNA-based LPS stimulation lasted for 1 hr, the protein levels of NF-κB phosphorylation were significantly increased compared to that of the group without IL-33 knockout. However, the phosphorylation of p38 was not significantly different within 30 min of LPS stimulus but elevated sharply when the stimulus lasted for

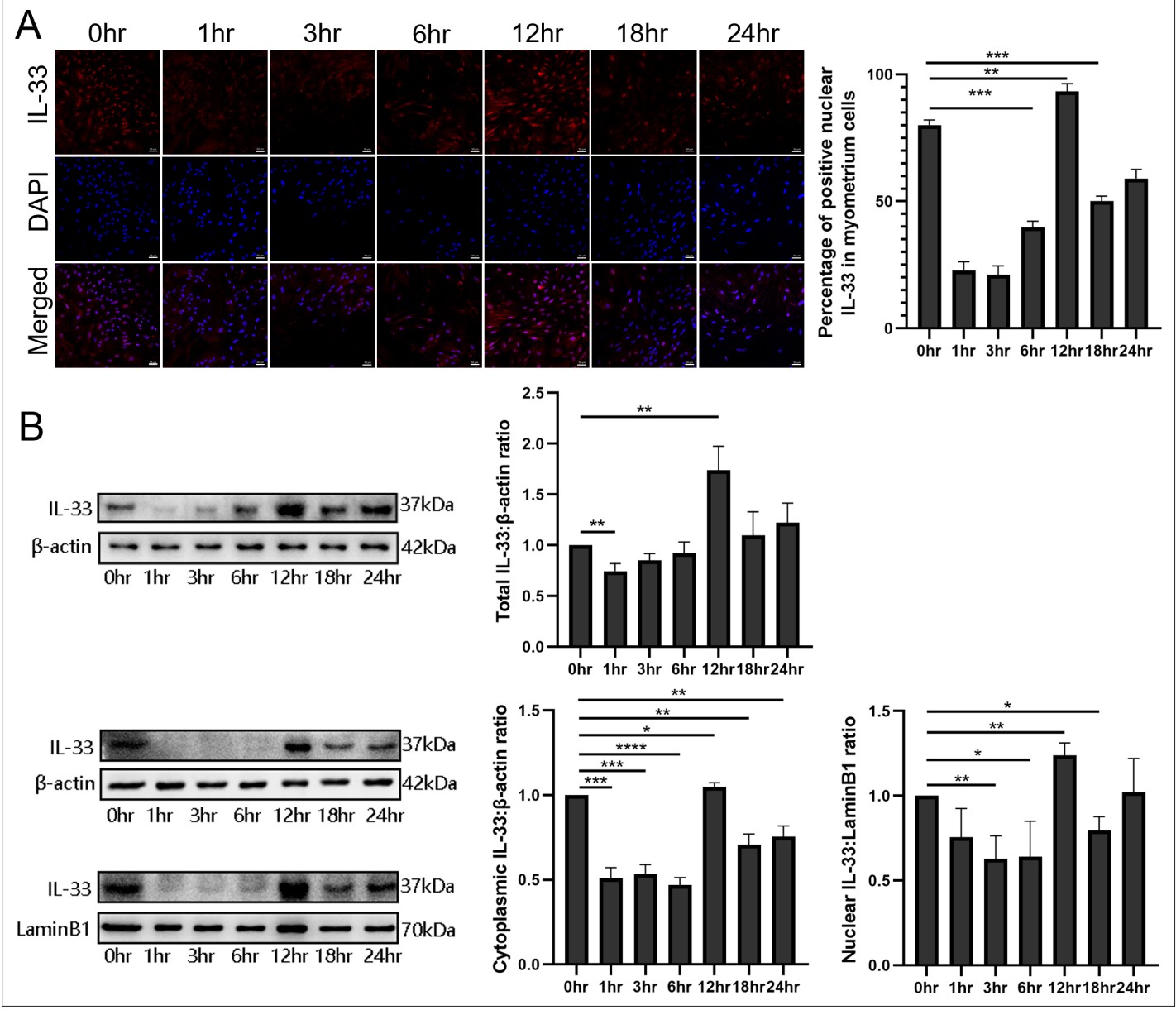

**Figure 2.** Dynamic changes in the localization of interleukin-33 (IL-33) during lipopolysaccharide (LPS) stimulation. (**A**) Localization analysis using confocal laser scanning microscopy images of IL-33 with the stimulation of 10 μg/ml LPS, IL-33 can be seen in the nucleus and this declines sharply compared with the control group, especially after 3 hr. However, with the longer time of the LPS treatment, the expression of IL-33 rises again from 6 hr and reaches peak at 12 hr. The lower panel is the percentage of cells of whereby IL-33 is expressed mainly in the nuclei (*n* = 3). Representative immunofluorescence photomicrographs are shown. (**B**) After the primary cells were loaded with LPS for different lengths of time, the cytoplasmic and nuclear proteins were isolated. In the cytoplasmic and nucleus fractions, IL-33 was quantified by western blotting, and the experiment was performed as described for (**B**). Each experiment was performed minimum of three times and the three representative results shown (*n* = 3). Representative blots are also shown. From the blots, we can see that the cytoplasmic and nuclear fraction IL-33 levels presented the same trend changes as the immunofluorescence data. The data from western blots and immunofluorescence are presented as means ± standard error of the mean (SEM) of the ratios. Statistical analyses were compared using an unpaired two-tailed Student's *t* test. *p＜0.05, **p＜0.01, *** P＜0.001, **** P＜0.0001, when compared to control; bar, 50 μm.

The online version of this article includes the following source data for figure 2:

**Source data 1.** Dynamic changes in the localization of IL-33 during LPS stimulation.

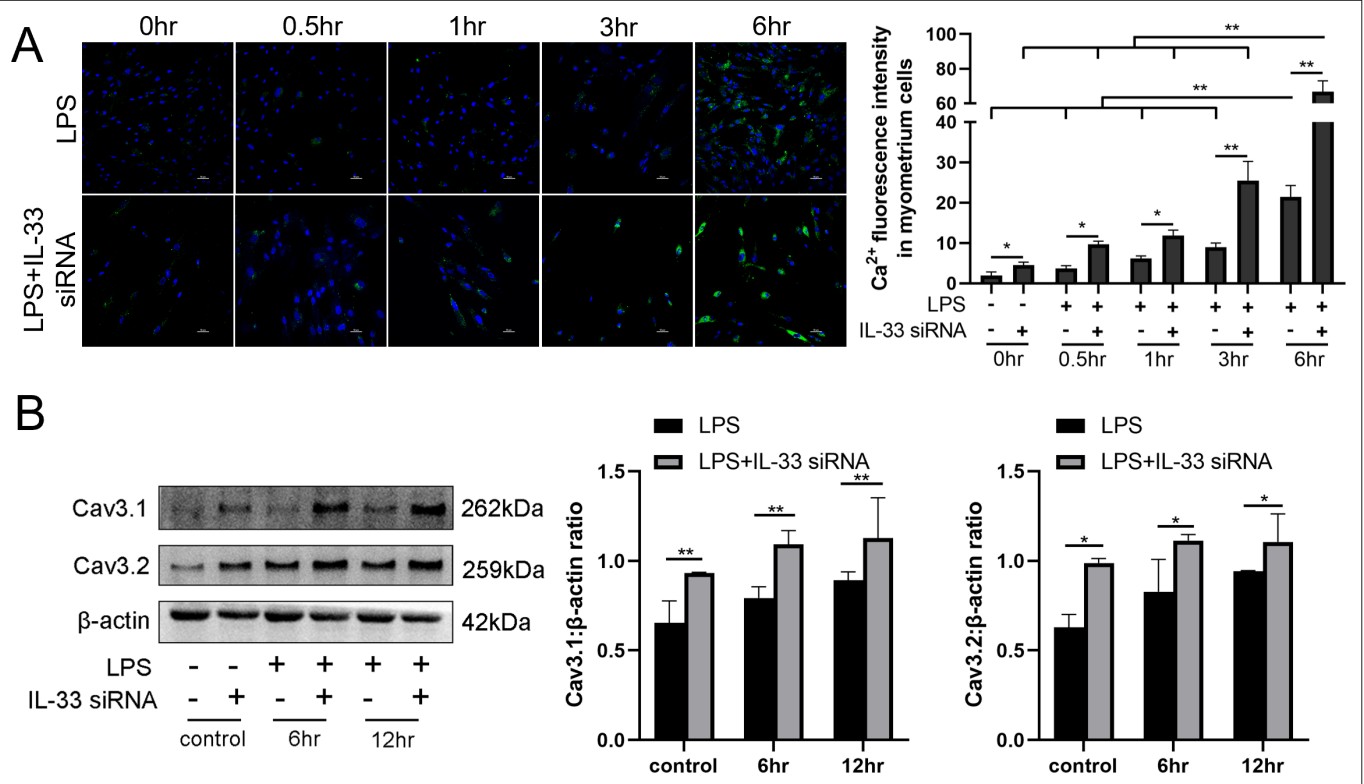

**Figure 3.** Interleukin-33 (IL-33) silencing enhanced lipopolysaccharide (LPS)-induced expression of calcium channels and the intracellular calcium concentration. Primary myometrium cells were transfected with IL-33 or nontargeting (as control) siRNAs and treated with 10 μg/ml LPS or phosphate-buffered saline (PBS) (as control) for 0.5, 1, 3, and 6 hr, respectively. (**A**) Immunofluorescence staining analysis was performed with fluo-3AM (green) and nuclei were stained with DAPI (blue) (*n* = 3). Representative immunofluorescence photomicrographs are shown. (**B**) Western blot analysis for the expression levels of Cav 3.1 and Cav 3.2 in stably transfected and nontransfected myometrial cells with the treatments as indicated (*n* = 5). Representative blots are shown. The data from western blots and immunofluorescence are presented as means ± standard error of the mean (SEM) of the ratios. Statistical analyses were compared using a one-way analysis of variance and Bonferroni multiple comparisons test. *p＜0.05, **p＜0.01 when compared to controls; bar, 50 μm.

The online version of this article includes the following source data for figure 3:

**Source data 1.** IL-33 silencing enhanced LPS-induced expression of calcium channels and the intracellular calcium concentration.

30 min (*Figure 7B*). In addition, p38 and NF-κB signaling pathway inhibitors, SB-202190 and JSH-23, respectively, were added before LPS treatment, and western blotting analysis showed that COX-2 expression was significantly reduced (*Figure 7C*).

## IL-33 siRNA and cytoplasmic calcium influenced the expression of IL-8 and IL-6 secreted by myometrial cells

The expression levels of IL-8 and IL-6 in the supernatants of the LPS and siRNA treated cells were increased when compared with the controls. In addition, in comparison with the LPS group, more IL-8 and IL-6 were released into the cell supernatants after IL-33 siRNA transfection and partial binding of cytoplasmic calcium ions (*Figure 8*).

## Discussion

Although the specific mechanism of premature delivery is still unclear, inflammation is currently recognized as one of the major causes of this phenomenon (*Hantoushzadeh et al., 2020*; *Gilman-Sachs et al., 2018*). Our current understanding of this process is that as the myometrium switches from a quiescent to a contractile state, it also undergoes a shift in signaling from anti-inflammatory to proinflammatory pathways (*Hadley et al., 2018*; *Shynlova et al., 2020*). As an inflammatory factor, IL-33 has many similar effects to IL-1β. IL-33 is composed of a N-terminal nuclear domain and a C-terminal

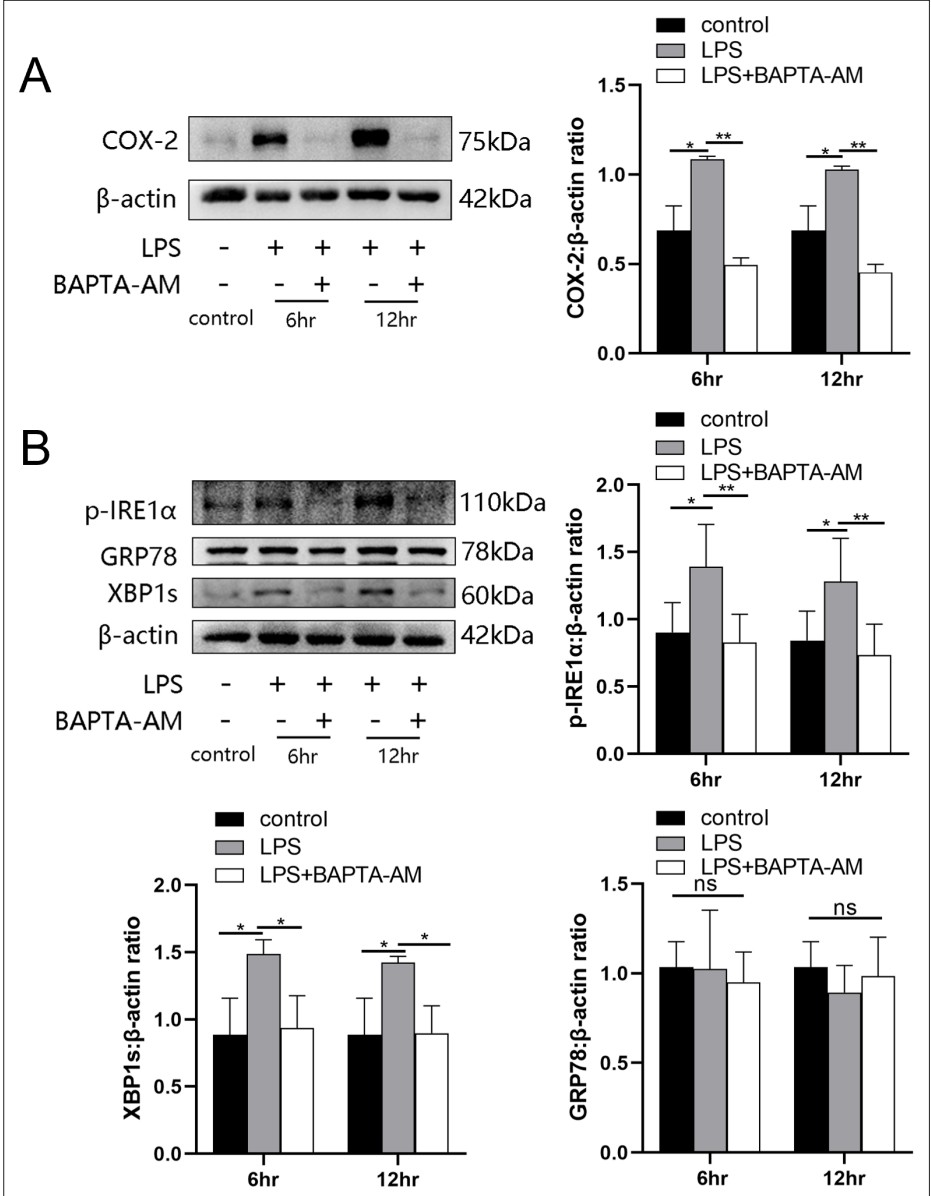

**Figure 4.** BAPTA-AM inhibited the expression of cyclooxygenase-2 (COX-2) and the endoplasmic reticulum (ER) stress response. (**A**) Study of COX-2 protein expression with lipopolysaccharide (LPS) treatment in the absence or presence of BAPTA-AM (25 μg/ml) (*n* = 5). (**B**) The protein expression of p-IRE1α, XBP1s, and GRP78 was analyzed by western blotting after 6 and 12 hr LPS stimulate and with or without BAPTA-AM (*n* = 5). Representative blots are shown for both (**A**) and (**B**). The data from western blots are presented as means ± standard error of the mean (SEM) of the ratios. Statistical analyses were compared using a one-way analysis of variance and Bonferroni multiple comparisons test. *p＜0.05, **p＜0.01 when compared to controls.

The online version of this article includes the following source data for figure 4:

**Source data 1.** BAPTA-AM inhibited the expression of COX-2 and the ER stress response.

IL-1-like cytokine domain which is mainly localized in the nucleus and binds to histones and chromatin (*Martin and Martin, 2016*). We proposed that IL-33 plays a dual role, playing a proinflammatory role as an inflammatory cytokine outside the cell as well as participating in the regulation of transcription as a nuclear factor (*Carriere et al., 2007*).

Numerous studies have shown that IL-33 had an essential role in pregnancy-related diseases such as recurrent abortion and preeclampsia (*Hu et al., 2015*; *Fock et al., 2013*). In early pregnancy, IL-33 can increase the proliferation and invasion of the decidua, and decidual macrophage-derived IL-33 is a key

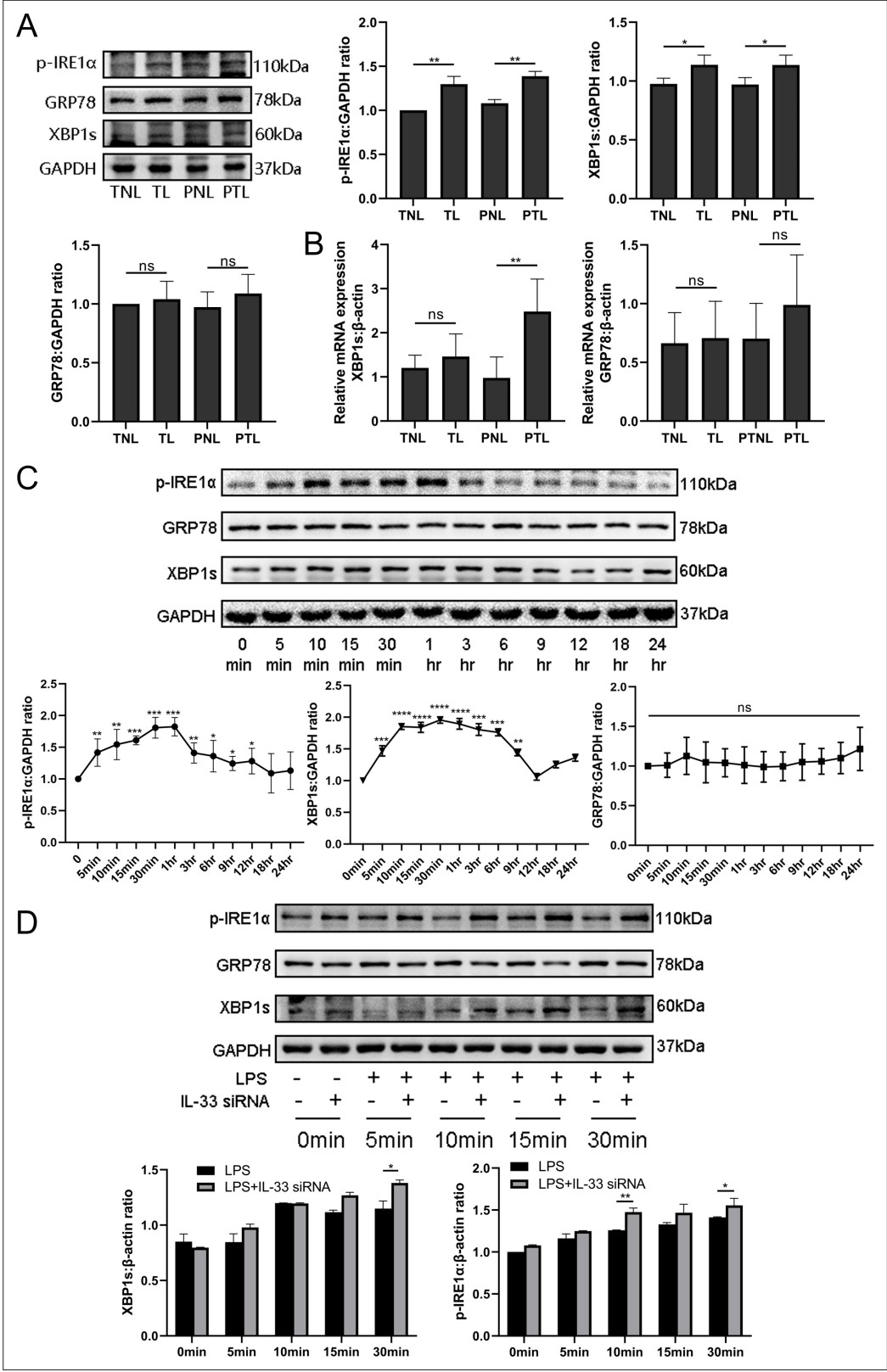

**Figure 5.** Interleukin-33 (IL-33) silencing highlighted the lipopolysaccharide (LPS)-induced endoplasmic reticulum (ER) stress response. Based on the discovery that calcium ions affected ER stress, western blotting and quantitative real-time PCR (qRT-PCR) were performed to explore the expression of ER stress in tissues at the protein and mRNA levels. (**A**) The protein levels of P-IRE1α and XBP1s in the term labor (TL) and preterm labor (PTL) groups were

*Figure 5 continued on next page*

*Figure 5 continued*

higher than those in the term nonlabor (TNL) and preterm nonlabor (PNL) groups while there was no alteration in the levels of GRP78 protein (*n* = 6). (**B**) The mRNA level of XBP1s in the PTL groups was higher than that in the PNL groups (*n* = 6). Furthermore, in order to illustrate the protein-level changes of ER stress during labor, LPS was used to stimulate primary uterine smooth muscle cells for different times and then western blot was used to assess the alteration of ER stress. (**C**) It was found that the protein levels of pIRE1α reached its peak at 10 min while XBP1s was at 15 min, while those of GRP78 did not change significantly (*n* = 5). We also detected apparent alterations in the ER stress protein when cells were stimulated based knockdown experiments targeting IL-33. (**D**) It revealed that the ER stress response in the siRNA-based group was more obvious compared with the LPS stimulated directly especially at 30 min (*n* = 5). For western blotting data with respect to (**A**), (**C**), and (**D**) representative blots are shown. The data are presented as mean percentages of control ± standard deviation (SD). The data from western blots are presented as means ± standard error of the mean (SEM) of the ratios. Statistical analyses were compared using an unpaired two-tailed Student's *t* test for (**A–C**), while a one-way analysis of variance and Bonferroni multiple comparisons test for (**D**). *$p < 0.05$, **$p < 0.01$, ***$p < 0.001$, ****$p < 0.001$ when compared to controls.

The online version of this article includes the following source data for figure 5:

**Source data 1.** IL-33 silencing highlighted the LPS-induced endoplasmic reticulum stress response.

factor for placental growth, and its deletion can lead to adverse pregnancy outcomes (*Sheng et al., 2018*; *Hu et al., 2014*). Similarly, IL-33 expression is reduced in the placenta of preeclamptic patients, which is a condition that affects placental function by reducing the proliferation, migration, and invasion of trophoblast cells (*Chen et al., 2018*; *Spaans et al., 2014*). In preliminary experiments, we found that IL-33 was expressed in the nuclei of myometrial cells in the third trimester of pregnancy and the levels decrease following the onset of the labor. This was consistent with previous studies which showed that IL-33, a mucosal alarmin, was being sequestered in the nucleus in order to limit its proinflammatory potential. It was released as a warning signal when tissues were damaged following stress or infection such as its release from the nucleus during asthma attack (*Kaur et al., 2015*; *Pelletier and Savignac, 2018*). However, IL-33 is mainly located in the nucleus of human myometrial tissue at the last trimester of pregnancy, but its functions remained unclear. Meanwhile, it was reported that IL-33 is a dual-function protein that might act both as a cytokine and as an intracellular nuclear factor (*Gautier et al., 2016*). In the following experiments, we explored the role(s) of the nuclear localization of IL-33.

In order to identify factors that are important in driving PTL progression, we utilized LPS to stimulate primary cultures of uterine smooth muscle cells to mimic inflammation during labor (*Sheller-Miller et al., 2021*; *Kent et al., 2022*). This resulted in a rapid decrease of IL-33 in the nucleus after LPS treatment which was ameliorated after the stimulus duration was lengthened. It is known that labor is a physiological process and will not cause a maternal pathological state, so we hypothesized that the recruitment of IL-33 may be to maintain cell homeostasis and enable its normal function. For exploring the potential effect of nuclear IL-33 during labor, we knocked down IL-33 in primary myometrial cells before LPS stimulation. Initially, we found that after introduction of siRNA IL-33, the intracellular calcium ion levels and the expression of T-type calcium channel proteins (Cav3.1 and Cav3.2) on the membrane, increased sharply. This was consistent with other studies which showed that IL-33 can affect calcium elevation autonomously or it can cooperate with other mediators to achieve this effect, although the specific mechanism is still unclear (*Gupta et al., 2017*). In addition, we tried to explain how IL-33 affected calcium ion levels, directly or via other subcellular organelles that regulated local calcium concentrations.

Myometrial contractions are mainly regulated by intracellular $Ca^{2+}$ levels and calmodulin and the effect of IL-33 on calcium elevation is likely to be a direct effect upon myometrial tissues (*Wray and Prendergast, 2019*). However, we found that reduced levels of IL-33 in the nucleus promoted the release of proinflammatory cytokines such as IL-8 and IL-6 in myometrial cells, while the expression of proinflammatory cytokines was reduced after $Ca^{2+}$ binding in the cytoplasm. This led us to conclude that IL-33 may play a certain role in the process of parturition. Intracellular calcium was closely linked to the ER stress response. It is well documented that when tissues and cells are irretrievably stimulated or damaged, an ER stress response occurs to maintain cell homeostasis. The ER, which is the main intracellular calcium storage chamber, plays a vital impact on maintaining $Ca^{2+}$ homeostasis in various organelles, and homeostasis of the ER is closely related to intracellular $Ca^{2+}$ concentration (*Almanza et al., 2019*). So far, we attempted to show that IL-33 is involved in onset of labor via ER stress which

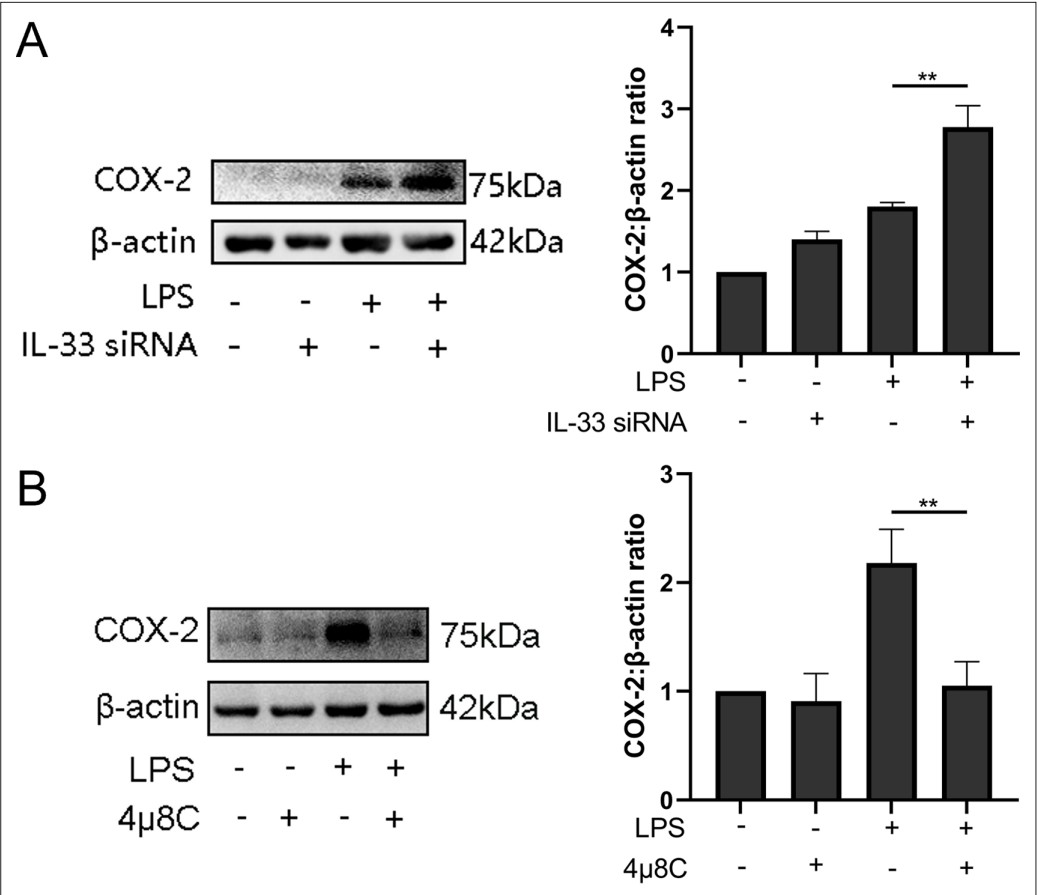

**Figure 6.** Interleukin-33 (IL-33) siRNA and the endoplasmic reticulum stress response affected cyclooxygenase-2 (COX-2) expression in myometrial cells. (**A**) In the process of studying whether IL-33 affects COX-2, we found that the COX-2 expression in the siRNA-mediated group was significantly increased compared with the lipopolysaccharide (LPS) alone group ($n = 5$). (**B**) Western blot analyses showing protein expression of COX-2 in myometrium cells was decreased following treatment with LPS for 12 hr ($n = 5$). Representative blots are shown for figures (**A**) and (**B**). The data from western blots are presented as means ± standard error of the mean (SEM) of the ratios. Statistical analyses were compared using an unpaired two-tailed Student's $t$ test. **p<0.01 when compared to controls.

The online version of this article includes the following source data for figure 6:

**Source data 1.** IL-33 siRNA and the endoplasmic reticulum stress response affected COX-2 expression in myometrial cells.

subsequently stimulates the release of calcium in the ER, thus leading the changes of in intracellular calcium concentrations and cell membrane calcium channels.

Researchers recently found that LPS stimulation of uterine explants could cause an ER stress response in uterine muscle. This led them to hypothesized that dysregulation of ER homeostasis in uterine smooth muscle might be one of the mechanisms of LPS-induced inflammatory preterm delivery (**Guzel et al., 2017**). When compared with nonlaboring specimens, the levels of GRP78, IRE1, and XBP1s in fetal membranes and myometrium were elevated in both TL and PTL (**Liong and Lappas, 2014**). Consistently, the expression of the ER stress response in the uterine smooth muscle was increased during labor when compared to nonlabor tissues. Here, we found that, although there was no obvious change in GRP78, and knockdown of IL-33 further increased LPS-induced expression of IRE1α and XBP1s, significantly. As a chaperone protein in the ER, GRP78 binds to three prominent ER stress proteins under steady-state conditions. These are PERK and ATF6, which are related to redox reaction and the Golgi apparatus, respectively. An additional protein, IRE1α, is also involved and it is related to inflammatory signals and the repair of misfolded proteins (**Ron and Walter, 2007**; **Oakes and Papa, 2015**). During the adaptive phase of the ER stress response, the IRE1α–XBP1s

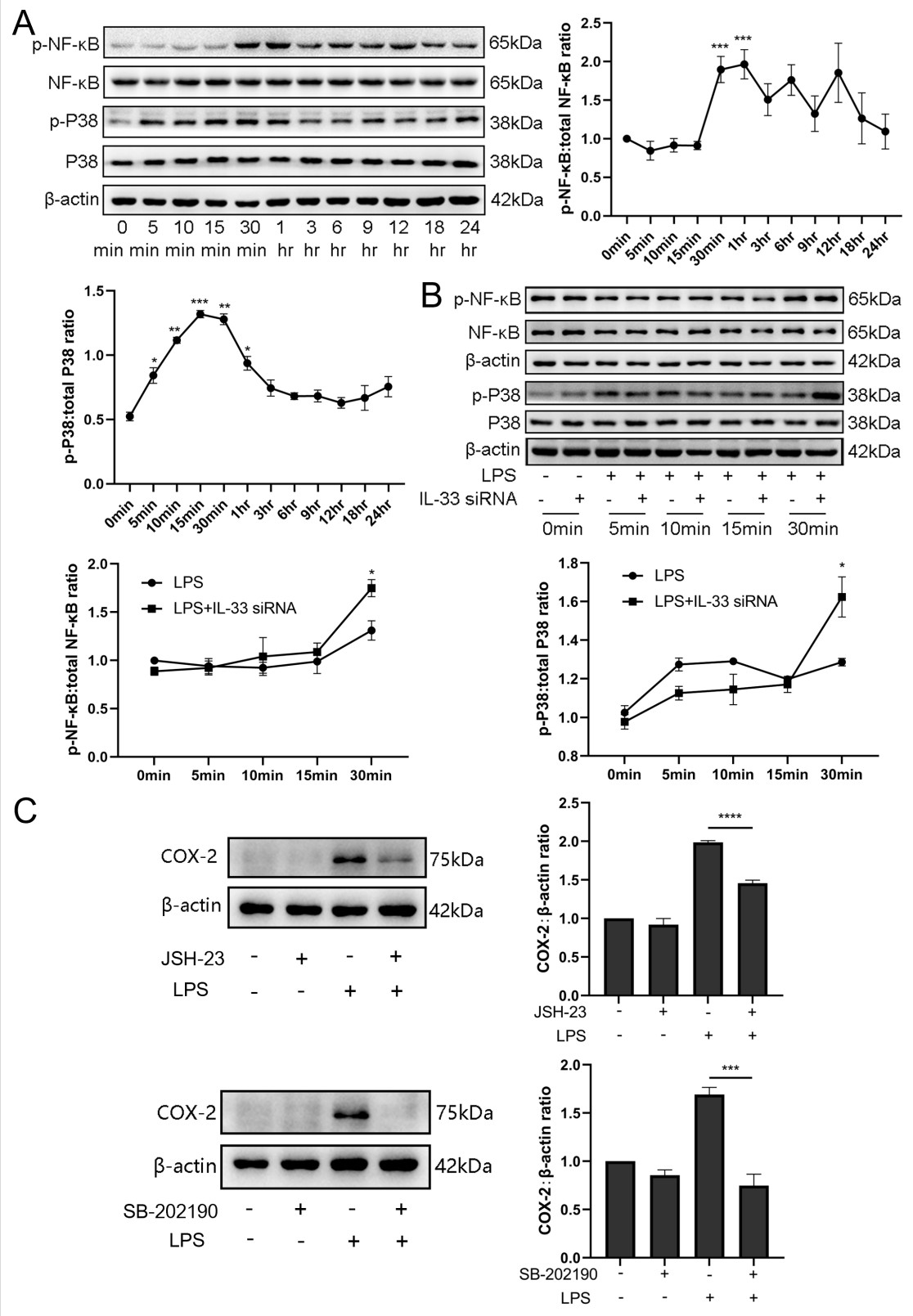

**Figure 7.** Interleukin-33 (IL-33) knockdown enhanced lipopolysaccharide (LPS)-induced nuclear factor kappa-B (NF-κB) and p38/mitogen-activated protein kinase (MAPK) signaling pathways. (**A**) Relative levels of p-P38, P38, p-NF-κB, and NF-κB were assessed by western blot analysis at the indicated time point after LPS (10 μg/ml) stimulation. Phosphorylation levels of P38 and NF-κB increased gradually with LPS stimulation and peaked at 15 min and 1 hr, respectively (n = 5). (**B**) Western blot analysis of p-P38, P38, p-NF-κB, and NF-κB expression in cells transfected with siRNA targeting

*Figure 7 continued on next page*

*Figure 7 continued*

IL-33 after treatment with LPS for 30 min. Compared with the LPS group, the protein expression of phosphorylated P38 and NF-κB were increased in the LPS + siRNA IL-33 group (*n* = 5). (**C**) The protein levels of cyclooxygenase-2 (COX-2) were decreased when SB-202190 and JSH-23 blocked the p38/MAPK and NF-κB signaling pathways, respectively (*n* = 5). Representative blots are shown for **A–C**. The data from western blots are presented as means ± standard error of the mean (SEM) of the ratios. Statistical analyses were compared using an unpaired two-tailed Student's *t* test for **A and C**, while a one-way analysis of variance and Bonferroni multiple comparisons test for **B**. *p<0.05, **p<0.01, ***p<0.001, ****P<0.0001, when compared to controls.

The online version of this article includes the following source data for figure 7:

**Source data 1.** IL-33 knockdown enhanced LPS-induced NF-κB and p38/MAPK signaling pathways.

pathway is activated. In the near future, we would like to confirm whether Ca Release Activated Ca (CRAC) is involved in ER stress response in order to maintain myometrial cell calcium homeostasis (*Zhang et al., 2020*).

To explore the potential mechanism of how IL-33 is involved in the process of labor, we transfected siRNA IL-33 into primary myometrial cells, and the decrease in nuclear IL-33 prompted the increase of COX-2 expression after LPS stimulation. Meanwhile, the stimulating effect of LPS on COX-2 which could be blocked by calcium chelating agents was more obvious. Furthermore, we explored whether the modest ER stress response seen relates to the process of labor. We found that the expression of COX-2 was sharply decreased by loading the primary cells with an ER stress response blocker which is similar with research results seen with modulation of pain by leukocytes (*Chopra et al., 2019*).

Based on the above findings, we speculated that IL-33 could influence LPS induced COX-2 expression via two pathways: (1) it directly prevents excessive COX-2 expression in myometrial cells after LPS stimulation and (2) it influences COX-2 expression by maintaining the severity of ER stress response. In addition, we also speculate that IL-33 may affect the myometrial cells calcium homeostasis via ER calcium-level regulation. The binding of IL-33 to NF-κB in the nucleus could block NF-κB signaling and inhibit the transduction of downstream associated inflammatory signals (*Griesenauer and Paczesny, 2017*). Consistent with previous studies (*Ali et al., 2011*), phosphorylation of NF-κB following LPS treatment was increased after IL-33 knockdown in myometrial cells. Also, we found that a change of the p38/MAPK signaling pathway was consistent with that of the NF-κB signaling pathway. In addition, selective blocking of NF-κB and p38/MAPK signaling pathways significantly inhibited the LPS-induced COX-2 expression. Therefore, we hypothesized that NF-κB and p38/MAPK signaling pathways may be the potential downstream pathways of IL-33 affecting LPS-induced COX-2 production (*Figure 9*).

To summarize, we found that the nuclear factor, IL-33, can affect contraction of uterine smooth muscle cells so as to participate in parturition by regulating levels of intracellular calcium ions and thus influencing the ER stress. Our study demonstrates that nuclear IL-33 in the myometrium participates in maintaining a uterine quiescent state at the tissue-to-cellular level during late pregnancy. Further

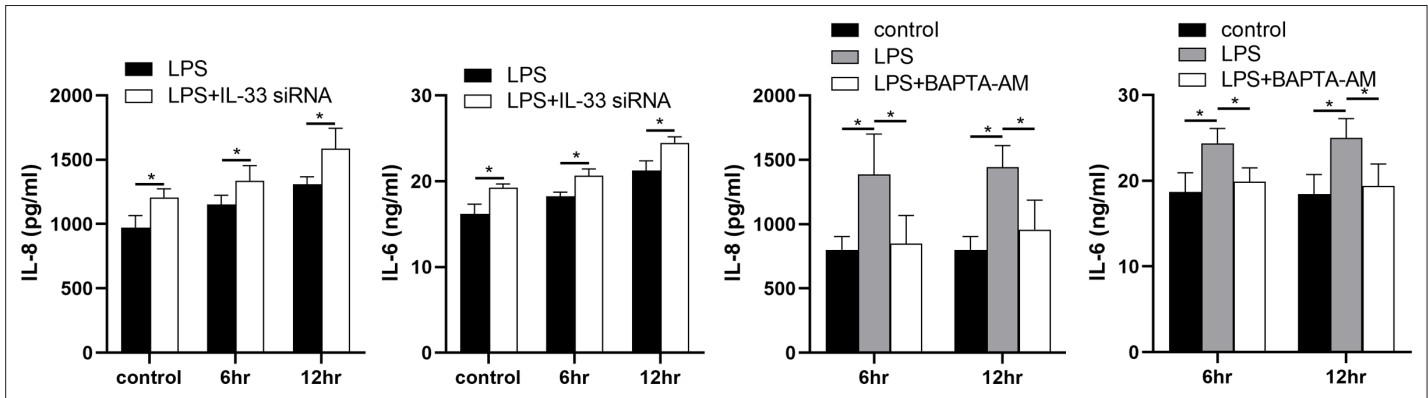

**Figure 8.** Interleukin-33 (IL-33) siRNA and the cytoplasmic calcium influenced the expression of IL-8 and IL-6. Compared with the lipopolysaccharide (LPS) group, the expression of IL-8 and IL-6 was increased in the LPS + IL-33 siRNA group and decreased in the LPS + BAPTA AM group (*n* = 5). The data are presented as mean percentages of control ± standard deviation (SD). Statistical analyses were compared using a one-way analysis of variance and Bonferroni multiple comparisons test. *p<0.05 when compared to controls.

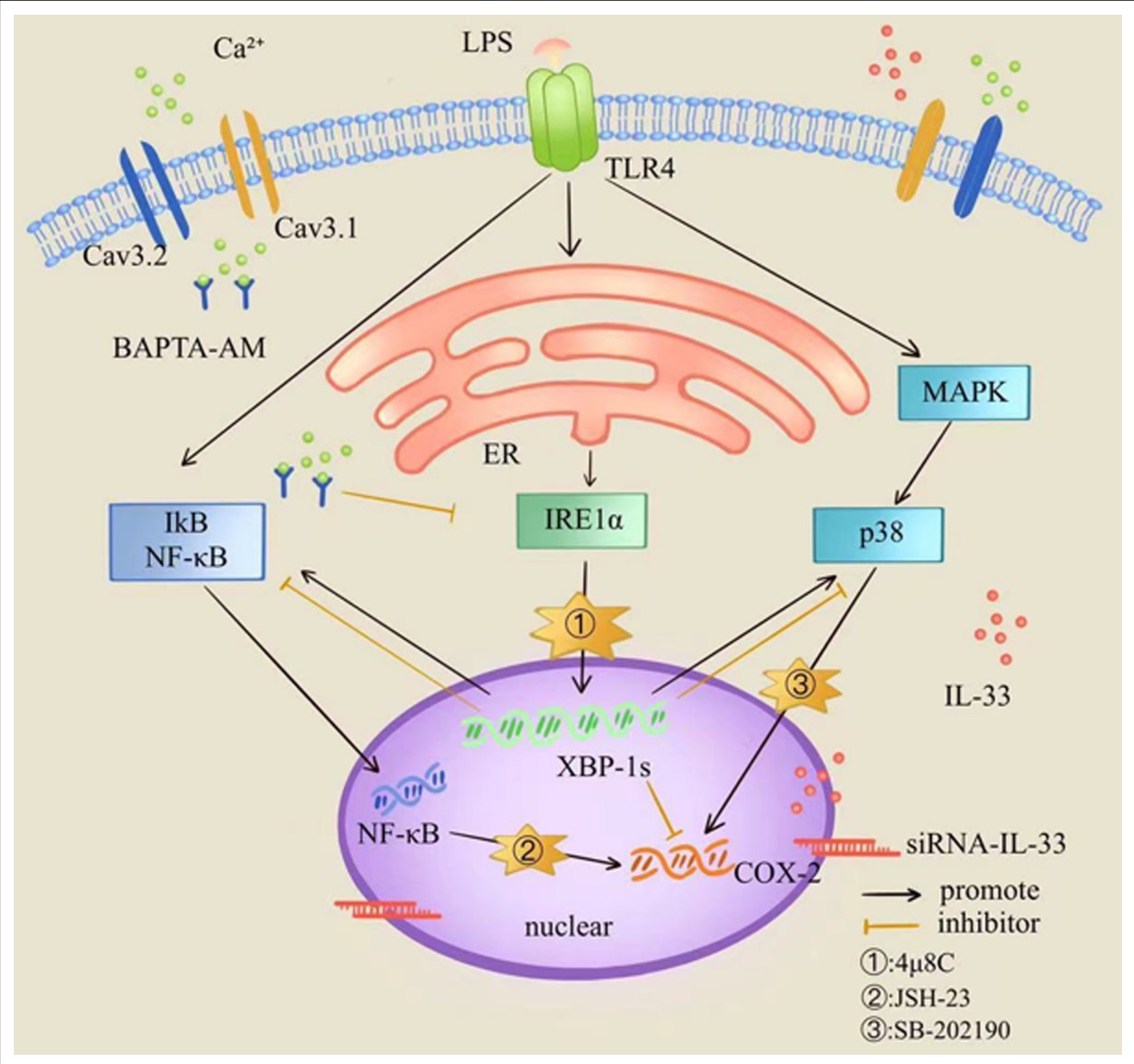

**Figure 9.** Diagrammatic model for the role of interleukin-33 (IL-33) in the myometrium where it participates in maintaining a uterine quiescent state at the tissue-to-cellular level during late pregnancy.

studies will be focused on underlying mechanism(s) whereby of nuclear IL-33 maintains myometrial calcium homeostasis via ER stress.

All data generated or analyzed during this study are included in the manuscript and supporting files.

## Acknowledgements

The authors thank Hong Zhou for advice and technical assistance in early of this work. This study was supported by grants from the National Natural Science Foundation of China (No. 81300507). The authors also thank Dr Dev Sooranna of Imperial College London for editing the manuscript.

## Additional information

### Funding

| Funder | Grant reference number | Author |
|---|---|---|
| National Natural Science Foundation of China | 81300507 | Li Chen |

The funders had no role in study design, data collection, and interpretation, or the decision to submit the work for publication.

### Author contributions

Li Chen, Conceptualization, Formal analysis, Supervision, Funding acquisition, Investigation, Writing – original draft, Project administration, Writing – review and editing; Zhenzhen Song, Resources, Data curation, Formal analysis, Validation, Investigation, Writing – original draft; Xiaowan Cao, Data curation, Formal analysis, Validation, Investigation, Methodology; Mingsong Fan, Resources, Software, Formal analysis; Yan Zhou, Software, Methodology; Guoying Zhang, Conceptualization, Supervision, Project administration, Writing – review and editing

### Author ORCIDs

Li Chen ⓘ http://orcid.org/0000-0002-6673-2963

### Ethics

All the patients signed an informed consent form. The study was approved by the Ethics Committee of the First Affiliated Hospital of Nanjing Medical University (2019-SRFA-159), China, and complies with the Declaration of Principles of Helsinki (2013).

### Decision letter and Author response

Decision letter https://doi.org/10.7554/eLife.75072.sa1
Author response https://doi.org/10.7554/eLife.75072.sa2

## Additional files

### Supplementary files

- Transparent reporting form
- Reporting standard 1. STROBE checklist.

### Data availability

All data generated or analyzed during this study are included in the manuscript and supporting files.

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
