## [Editor Report]

This paper addresses an interesting and important topic bearing on the initiation of labor at the end of pregnancy. It tests a hypothesis invoking interleukin-33 in uterine smooth muscle in the third trimester of pregnancy in an endoplasmic reticular stress. This in turn results in an alteration of ca^2+^ homeostasis that might be involved in initiating labor. The study was made in human myometrial cells enhancing its clinical translatability.

---

## [Decision Letter]

**Decision letter after peer review:**

Thank you for submitting your article "Interleukin-33 stimulates stress in the endoplasmic reticulum of the human myometrium via an influx of calcium during initiation of labor" for consideration by *eLife*. Your article has been reviewed by 3 peer reviewers, one of whom is a member of our Board of Reviewing Editors, and the evaluation has been overseen Mone Zaidi as the Senior Editor. The reviewers have opted to remain anonymous.

The reviewers have discussed their reviews with one another, and the Reviewing Editor has drafted this to help you prepare a revised submission. The dialog between all the reviewers 1,2 and 3 emerges with a consensus that the paper addresses an interesting an important topic bearing on the initiation of labor at the end of pregnancy, invoking interleukin-33 in an alteration of ca^2+^ homeostasis in uterine smooth muscle. However, they point out aspects in which the evidence falls short of the conclusion you wish to draw. Nevertheless consultation between editors lead us to wish to give you an opportunity to make major revisions that involve additional experiments for a revised version, which we list below. In the latter connection, the list of essential revisions below refer both to the additional evidence required, as well as refer to the list of corrections under the headings of 'recommendations for the authors' to assist such a revision. We do this with the expectation that these are feasible within a 6 month period.

Essential revisions:

(1) In addition to the changes requested in the notes below, reviewer 2 flags that the conclusions drawn here would require a good amount of additional work to fully support.

(2) Reviewer 3 makes parallel criticisms, considering that much stronger data is needed to make a strong case worthy of consideration for publication to *eLife*, particularly of IL33 function in the rodent model of LPS-induced preterm labour.LPS as a TLR4 activator will drive expression of multiple cytokines some via NFkB and also other pro-contractile molecules. Thus both reviewers felt the characterisation of the responses and role of IL33 inadequate to make a strong case for this cytokine.

(3) Reviewer 2 under "recommendations for the authors" make comments that require addressing concerning the data in its present form that require addressing.

(4) Reviewer 3 also refers to a need of studies on primary tissues (not cells which tend to lose their contractile phenotype once cultured) to back up their claim that this is an important molecule involved in the onset of labour.

*Reviewer #1 (Recommendations for the authors):*

Matters of clarity.

(1) Abbreviations need to be defined with their use in the abstract.

(2) Make more explicit the experimental groups:

a) term labor (TL)

b) preterm labor (PTL)

c) nonlabor (TNL)

d) nonlabor (PNL)

(3) The relationship between the LPS, the silencing and the ER stress experiments needs to be made more explicit, as at present the paper appears disjointed, with the reader tending to lose the plot.

*Reviewer #2 (Recommendations for the authors):*

– Figure 1 legend line 110 to say QRTPCR rather than QTPCR.

– Nuclear cytosolic protein westerns should show any presence of cross contamination. Thus, any β actin contamination in nuclear fraction should be shown as well as lamin B contamination in cytosolic fraction.

– In general, Figure legends need details of statistics used.

– Different LPS serotypes was shown to activate different inflammatory pathways. Please clarify which serotype was used.

– Calcium response for contractions is often a quick cyclic response. Would be interesting to see the timepoints shorter than 6h? potentially measuring calcium response real time using fluo4AM would be informative in linking to uterine contractions. Can IL-33 induce calcium response alone?

– Control should be included in all IL-33 siRNA experiments to demonstrate successful knockdown of IL-33.

– Figure 4B shows LPS increasing p-IRE1alpha and XBP1s at 6h but this is not shown in Figure 5C during a timecourse experiment (only XBP1s increased). Can you explain what causes this difference?

– NL vs L for ER stress markers. Protein difference is apparent but not mRNA. This might be due to missed time point as mRNA can increase before protein levels. This should be discussed.

– Figure 5C and 5D with LPS only doesn't show same results. Was significant increase in p-IRE1alpha seen with LPS alone in Figure 5D? if not, why was this different to 5C.

– Figure 6 shows LPS-induced COX2 expression involves ER stress, not IL-33. If authors want to state that the enhanced LPS-induced COX2 expression via knockdown of IL-33 involves ER stress, they should perform experiment using the ER stress blocker in IL-33 knockdown cells following LPS treatment.

– Figure 7B, why was timecourse done for p38 but not NFkappaB?

– Figure 7C shows LPS-induced COX2 expression is via p38 and NFkappaB, not IL-33. This needs to be clarified in the manuscript.

– Figure 8 shows IL-33 siRNA affecting basal level of IL-8 with no LPS, this should be discussed. Also, it shoes LPS-induced calcium response plays a role in IL-8 and IL-6 levels, not IL-33. This should be made clear.

– Discussion line 300 'ta previous study' should be 'the previous study'.

– Authors should clarify that inflammatory response induced by LPS is infection-induced, which is not necessarily the same as those induced by other stimulants of labor.

– Discussion line 313-315, authors state that they found that after introduction of siRNA IL-33, the intracellular calcium ion levels and the expression of T-type calcium channel proteins (Cav3.1 and Cav3.2) on the membrane increased sharply. However, siRNA IL-33 effect on calcium levels was not shown (what was shown was siRNA IL33 effect on LPS-induced calcium response) only that of calcium channel proteins.

– Discussion line 321-324, authors state that they found that reduced IL-33 in the nucleus promoted the release of pro-inflammatory cytokines such as IL-8 and IL-6 in myometrial cells, while the expression of pro-inflammatory cytokines was reduced after ca^2+^ binding in the cytoplasm. The data presented only shows that IL33 knockdown alone induced release of IL-8 not IL-6. It enhanced release of LPS-induced IL-8 and IL-6, and calcium chelation was able to reduce LPS-effect, not that of IL-33.

– Discussion line 338-340, authors state that knock down of IL-33 increased the expression of IRE1α and XBP1s significantly. It should be clarified that knockdown of IL33 enhanced LPS-induced expression of IRE1alpha and XBP1s, not IL33 alone.

– Discussion line 352-353, 357-358, authors state that IL-33 influenced COX-2 expression by maintaining the severity of ER stress response. However, it should be clarified that the LPS-induced expression of COX2 was decreased by blocking ER stress response, not IL-33 as this experiment was not shown in the manuscript.

– Discussion line 373-375, authors conclude that nuclear IL-33 in the myometrium participates in maintaining a uterine quiescent state at the tissue-to-cellular level during late pregnancy. Further studies on whether IL-33 is involved in the redox reaction of mitochondria as well as regulation of intracellular calcium ions caused by ER stress are necessary. The study indicates low levels of IL-33 can enhance LPS/infection-induced inflammation during late pregnancy. As no experiments were done using over-expression of IL-33, it does not necessarily demonstrate that IL-33 maintains uterine quiescence.

– Authors should also indicate gestational age window for myometrial samples for all groups.

– Methods line 457 subscript bracket to be edited.

– The siRNA concentrations used should be added to methods, and clarify appropriate vehicle controls were used for BAPTA-AM and 4μ8C.

*Reviewer #3 (Recommendations for the authors):*

As mentioned previously, the manuscript would be of some interest to cell biologists examining pathways of myometrial contractility but has very limited, if any, translational potential.

The authors need to strengthen the conceptual aspect of myometrial IL33 in relation to other inflammatory mediators and also carry out extensive studies on primary tissues (not cells which tend to lose their contractile phenotype once cultured) to back up their claim that this is an important molecule involved in the onset of labour.

---

## [Author Response]

Reviewer #1 (Recommendations for the authors):Matters of clarity.(1) Abbreviations need to be defined with their use in the abstract.

We have defined all of the abbreviations in the abstract in revised manuscript as the Reviewer suggests.

(2) Make more explicit the experimental groups:a) term labor (TL)

a) Term labor (TL): samples were obtained from women who had the presence of contractions of sufficient strength and frequency to effect progressive effacement and dilation of the cervix and were between 37 and 41+6 weeks' gestation.

b) preterm labor (PTL)

b) Preterm labor (PTL): samples were obtained from women who had the presence of contractions of sufficient strength and frequency to effect progressive effacement and dilation of the cervix and were between 28 and 36+6 weeks' gestation.

c) nonlabor (TNL)

c) Term non-labor (TNL): These samples were obtained from women who were between 37 and 41+6 weeks' gestation without any signs of onset of labor.

d) nonlabor (PNL)

d) preterm nonlabor (PNL): These samples were obtained from women who were between 28 and 36+6 weeks' gestation without any sign of onset of labor.

These were defined in the Materials and methods in revised manuscript.

(3) The relationship between the LPS, the silencing and the ER stress experiments needs to be made more explicit, as at present the paper appears disjointed, with the reader tending to lose the plot.

We have clarified the relationship between LPS and siRNA IL-33 and ER stress in the discussion of the revised manuscript as the Reviewer suggests.

Reviewer #2 (Recommendations for the authors):– Figure 1 legend line 110 to say QRTPCR rather than QTPCR.

We have corrected this as the Reviewer suggests.

– Nuclear cytosolic protein westerns should show any presence of cross contamination. Thus, any β actin contamination in nuclear fraction should be shown as well as lamin B contamination in cytosolic fraction.

We show the W-B here. We repeated the experiment three times and Author response image 1 is a representative blot.

**Author response image 1. sa2fig1:** 

– In general, Figure legends need details of statistics used.

These were explained in the Figure legends of the revised manuscript as the Reviewer suggests.

– Different LPS serotypes was shown to activate different inflammatory pathways. Please clarify which serotype was used.

We used the LPS serotype, O55:B5, which was based on previous studies that used LPS-treated cells in preterm labor.

– Calcium response for contractions is often a quick cyclic response. Would be interesting to see the timepoints shorter than 6h? potentially measuring calcium response real time using fluo4AM would be informative in linking to uterine contractions. Can IL-33 induce calcium response alone?

We have accomplished this part of the experiments and the new time course used were included 0.5, 1, 3 and 6 hours. When we knocked down IL-33, we found an increase in intracellular calcium levels (Figure 3 of the manuscript).

– Control should be included in all IL-33 siRNA experiments to demonstrate successful knockdown of IL-33.

We show in Author response image 2 the efficiency of siRNA was estimated to be at least 75%.

– Figure 4B shows LPS increasing p-IRE1alpha and XBP1s at 6h but this is not shown in Figure 5C during a timecourse experiment (only XBP1s increased). Can you explain what causes this difference?

We re-analyzed and also repeated the experiments two more time (Figure 5C data), and we found the same trend. In other words, LPS increased both p-IRE1alpha and XBP1s at 6h.

– NL vs L for ER stress markers. Protein difference is apparent but not mRNA. This might be due to missed time point as mRNA can increase before protein levels. This should be discussed.

For this part of the experiments, myometrium tissue samples were obtained from the pregnant women in labor and not in labor but without any time points. We considered post-transcriptional modification might be involved in ER stress for human onset of labor. In our future research, we hope to further confirm this hypothesis.

– Figure 5C and 5D with LPS only doesn't show same results. Was significant increase in p-IRE1alpha seen with LPS alone in Figure 5D? if not, why was this different to 5C.

We repeated this part of the work more three times, and analyzed the data again. When the primary culture cells were treated with LPS alone, the expression of p-IRE1alpha was also significantly increased.

– Figure 6 shows LPS-induced COX2 expression involves ER stress, not IL-33. If authors want to state that the enhanced LPS-induced COX2 expression via knockdown of IL-33 involves ER stress, they should perform experiment using the ER stress blocker in IL-33 knockdown cells following LPS treatment.

For this experiment, we only confirmed that both ER stress and IL-33 were involved in LPS-induced COX2 expression. As you can see, for the whole project, LPS was a basal treatment and we attempted to illustrate that IL-33 was involved in LPS induced inflammation. However, when we treated primary myometrial cultured cells with siRNA IL-33+LPS+ER inhibitor, we only observed a decrease in COX-2 expression, but we could not see any differences after double or triple treatments.

– Figure 7B, why was timecourse done for p38 but not NFkappaB?

We have add the data regarding time course done for NF-kappaB to Figure 7B as the Reviewer suggests.

– Figure 7C shows LPS-induced COX2 expression is via p38 and NFkappaB, not IL-33. This needs to be clarified in the manuscript.

The data confirms that LPS-induced COX2 expression occurs via p38 and NF-kappaB. As both ER stress and IL-33 are involved in LPS-induced COX2 expression, this further confirms that pathways involved**.** Please see our concluding sentences at the end of the Discussion in the revised manuscript.

– Figure 8 shows IL-33 siRNA affecting basal level of IL-8 with no LPS, this should be discussed. Also, it shoes LPS-induced calcium response plays a role in IL-8 and IL-6 levels, not IL-33. This should be made clear.

We repeated the experiments three more times, and re-analyzed the data and found that siRNA alone also can also increase IL-6 and IL-8 expression.

– Discussion line 300 'ta previous study' should be 'the previous study'.

We have corrected the sentence. Thank you for your kind suggestion.

– Authors should clarify that inflammatory response induced by LPS is infection-induced, which is not necessarily the same as those induced by other stimulants of labor.

We agree that LPS cannot represent inflammation-induced labor. In our previous work, we used both IL-1beta and LPS to treat cells. However, IL-1beta also belongs to the IL-1 family and is therefore similar to IL-33. So it could not be used here. Also, based on two recently published papers, we use LPS to mimic inflammation induced labor.

1) Sheller-Miller S, Radnaa E, Yoo JK, et al., Exosomal delivery of NF-κB inhibitor delays LPS-induced preterm birth and modulates fetal immune cell profile in mouse models. Sci Adv 2021 01;7(4)

2) Kent LN, Li Y, Wakle-Prabagaran M, et al., Blocking the BKCa channel induces NF-κB nuclear translocation by increasing nuclear calcium concentration. Biol Reprod 2022 03 19;106(3)

– Discussion line 313-315, authors state that they found that after introduction of siRNA IL-33, the intracellular calcium ion levels and the expression of T-type calcium channel proteins (Cav3.1 and Cav3.2) on the membrane increased sharply. However, siRNA IL-33 effect on calcium levels was not shown (what was shown was siRNA IL33 effect on LPS-induced calcium response) only that of calcium channel proteins.

We have add the siRNA IL-33 data to Figure 3A as the Reviewer suggests.

– Discussion line 321-324, authors state that they found that reduced IL-33 in the nucleus promoted the release of pro-inflammatory cytokines such as IL-8 and IL-6 in myometrial cells, while the expression of pro-inflammatory cytokines was reduced after ca^2+^ binding in the cytoplasm. The data presented only shows that IL33 knockdown alone induced release of IL-8 not IL-6. It enhanced release of LPS-induced IL-8 and IL-6, and calcium chelation was able to reduce LPS-effect, not that of IL-33.

We repeated this experiment and re-analyzed the data and we found that knockdown IL-33 alone also can induce IL-6 and IL-8 release.

– Discussion line 338-340, authors state that knock down of IL-33 increased the expression of IRE1α and XBP1s significantly. It should be clarified that knockdown of IL33 enhanced LPS-induced expression of IRE1alpha and XBP1s, not IL33 alone.

Thank you for your kind suggestion and we have corrected these sentences as the Reviewer suggests

– Discussion line 352-353, 357-358, authors state that IL-33 influenced COX-2 expression by maintaining the severity of ER stress response. However, it should be clarified that the LPS-induced expression of COX2 was decreased by blocking ER stress response, not IL-33 as this experiment was not shown in the manuscript.

We have corrected this part of the Discussion as the Reviewer suggests.

– Discussion line 373-375, authors conclude that nuclear IL-33 in the myometrium participates in maintaining a uterine quiescent state at the tissue-to-cellular level during late pregnancy. Further studies on whether IL-33 is involved in the redox reaction of mitochondria as well as regulation of intracellular calcium ions caused by ER stress are necessary. The study indicates low levels of IL-33 can enhance LPS/infection-induced inflammation during late pregnancy. As no experiments were done using over-expression of IL-33, it does not necessarily demonstrate that IL-33 maintains uterine quiescence.

In our present research, we focused on the role(s) of the nuclear localization of IL-33. As we know, without its receptor(s), IL-33 alone can also play a role in the nucleus. When we overexpressed IL-33, the absolute concentration of cytoplasmic IL-33 was increased, and it could bind to its receptor(s) to play its function. This makes it more complicated when trying to illustrate the mechanism(s) of this cytokine.

– Authors should also indicate gestational age window for myometrial samples for all groups.

These were defined in the Materials and methods in revised manuscript as mentioned in the response to Reviewer 1.

– Methods line 457 subscript bracket to be edited.

Thank you for your reminder. This was corrected in the revised manuscript.

– The siRNA concentrations used should be added to methods, and clarify appropriate vehicle controls were used for BAPTA-AM and 4μ8C.

We have added the siRNA concentrations used in the Methods section of the revised manuscript. PBS was used as vehicle controls for BAPTA-AM and 4μ8C.

Reviewer #3 (Recommendations for the authors):As mentioned previously, the manuscript would be of some interest to cell biologists examining pathways of myometrial contractility but has very limited, if any, translational potential.

Thank you for your very kind suggestions. We feel that increasing our knowledge of the role of IL-33 on human myometrium contractility and how nuclear IL-33 reduces ER stress in order to regulate calcium homeostasis are important mechanisms to help us understand the problems associated with high risk pregnancies.

The authors need to strengthen the conceptual aspect of myometrial IL33 in relation to other inflammatory mediators and also carry out extensive studies on primary tissues (not cells which tend to lose their contractile phenotype once cultured) to back up their claim that this is an important molecule involved in the onset of labour.

Actually, in future studies we want to stretch the myometrium tissues and then treated the samples with an IL-33 activator or inhibitor to investigate the changes of myometrium contractility. This is a part of our on-going commitment to further understand the complications of pregnancy. Unfortunately, restrictions due to COVID-19, have delayed our work. Meanwhile, we hope to expand the work by using a knockout IL-33 mouse model. This model was expected to cause preterm labor in the mice, but unfortunately, this cytokine affected the conception and live birth rates significantly. Further work is continuing along this avenue of research we hope to further define the role of IL-33 in pregnancy-related diseases.